# Impact of Powder Properties on the Rheological Behavior of Excipients

**DOI:** 10.3390/pharmaceutics13081198

**Published:** 2021-08-04

**Authors:** Pauline H. M. Janssen, Sébastien Depaifve, Aurélien Neveu, Filip Francqui, Bastiaan H. J. Dickhoff

**Affiliations:** 1DFE Pharma, Klever Str. 187, 47574 Goch, Germany; bastiaan.dickhoff@dfepharma.com; 2Granutools, Rue Jean-Lambert Defrêne, 107, 4340 Awans, Belgium; sebastien.depaifve@granutools.com (S.D.); aurelien.neveu@granutools.com (A.N.); filip.francqui@granutools.com (F.F.)

**Keywords:** powder flow, die filling, powder rheology, dynamic cohesive index, excipients, tableting, continuous manufacturing, quality by design

## Abstract

With the emergence of quality by design in the pharmaceutical industry, it becomes imperative to gain a deeper mechanistic understanding of factors impacting the flow of a formulation into tableting dies. Many flow characterization techniques are present, but so far only a few have shown to mimic the die filling process successfully. One of the challenges in mimicking the die filling process is the impact of rheological powder behavior as a result of differences in flow field in the feeding frame. In the current study, the rheological behavior was investigated for a wide range of excipients with a wide range of material properties. A new parameter for rheological behavior was introduced, which is a measure for the change in dynamic cohesive index upon changes in flow field. Particle size distribution was identified as a main contributing factor to the rheological behavior of powders. The presence of fines between larger particles turned out to reduce the rheological index, which the authors explain by improved particle separation at more dynamic flow fields. This study also revealed that obtained insights on rheological behavior can be used to optimize agitator settings in a tableting machine.

## 1. Introduction

Powders are used in many industries and in a broad range of processes and applications. Often understanding powder behavior is crucial to properly design the process and equipment [1]. In a continuous manufacturing line for example, consistent and continuous flow through the system is a critical requirement for finished product quality [2]. Flow is also relevant for manufacturing efficiency for batch processes. It determines for example whether bins can be used or hand scooping is required, to what extend product is scraped at the beginning or end of a run, and the allowable production rate of products [3].

In pharmaceutical processing, insufficient flow can lead to product quality failures, due to large weight or dosage variations [4]. Weight variations in the final dosage form can occur when the flowability of the final formulation limits the separation of small quantities of powder from the larger mass of powder. Because of powder flow limitations in this step, a slow-speed process that works well may not work at all when rates are increased [3]. The production capacity of a tablet press is directly determined by the rotation frequency of the die table and limited by powder flow into the dies [5,6].

Predicting how powder will flow into the dies is however not easy, as flowability is a complex, multidimensional property. Flowability is the result of material physical properties and the equipment used for handling, storing or processing the material [3]. The forces that influence material flowability depend on the flow field and stress state as well as on external factors, like equipment design and material, temperature, and humidity. This also explains why flow behavior (and ranking) of powders can be different between different measurement methods and applications.

Many different characterization techniques to measure flow have been developed and correlated to the powder flow behavior in different processing units [1,7]. Pharmaceutical compendial flow characterization methods include angle of repose [8,9], compressibility index [10,11], flow through an orifice [12], rotating drum [13,14], powder rheometers [15], and several variants of the shear cell tester [16]. The different flow characterization methods quantify powder flow differently, due to the differences in flow field and degree of stresses applied during measurement [17]. Stavrou et al. [18] for example showed differences in powder flow behavior when different stress levels were employed.

Although many different characterization techniques have been developed, care must be taken to select the most suitable characterization technique for a specific approach [1]. Simple flow measurements tend to fail in the prediction of powder flow into tableting dies, due to a lack of simulation of the right flow field and stress state. So far, only a few characterization techniques have been developed with specific focus to mimic the die filling process. Wu et al. [19] for example used transparent dies and moving feeding shoes to study the powder flow of different metallurgical powder components in air and vacuum. He showed that powder characteristics, shoe speed, and die geometry play an important role in the die filling process. Mendez et al. [20] used a fixed frame and a moving die disc system to examine the effect of blend composition, shoe properties and die parameters on uniformity of die filling. Mehrotra et al. [21] simulated the die filling process for cohesive materials. He concluded that cohesive powder take longer to fill dies and hence could be a potential cause of tablet weight variability.

Described research has shown to predict the tablet die filling process from a die shoe well. A common practical challenge of the described studies however, is that they all studied the die filling process with simplified systems. The effect of different flow fields in the feeding frame has not been considered. This factor can be very important in the prediction of die filling, especially when powder fluidity is influenced by the presence of paddle feeders, wipers, or agitator arms [3].

With the emergence of quality by design (QbD), it becomes imperative to gain a deeper mechanistic understanding of how different materials respond to differences in flow field [22]. The aim of this research is to investigate the effect of rheological behavior during powder flow into tablet dies. A wide range of excipients with a broad range of material properties was evaluated. A new parameter for rheological behavior is introduced, which is a measure for the change in dynamic cohesive index upon changes in stress state and flow field. The objective of this study is to identify which material properties do have an impact on this rheological index parameter. In addition, the fidelity of the rheological index was validated by correlating it to tableting performance in a rotary tablet press with agitators.

## 2. Materials and Methods

### 2.1. Materials

Anhydrous lactose (Lactopress^®^ anhydrous, SuperTab^®^ 21 AN, SuperTab^®^ 22 AN, SuperTab^®^ 24 AN), sieved lactose monohydrate (Pharmatose^®^ 80 M), milled lactose monohydrate (Pharmatose^®^ 150 M, Pharmatose^®^ 200 M, Pharmatose^®^ 450 M), modified lactose monohydrate (SuperTab^®^ 11 SD, SuperTab^®^ 14 SD, SuperTab^®^ 50 ODT, SuperTab^®^ 30 GR), microcrystalline cellulose (Pharmacel^®^ 101, Pharmacel^®^ 102), and superdisintegrants (Primojel^®^, Primellose^®^) were obtained from DFE Pharma (Goch, Germany).

### 2.2. Material Characterization

An overview of the characterization techniques, with corresponding physical properties and abbreviations is provided in Table 1.

#### 2.2.1. Shape

Scanning electron microscopy (SEM) images were recorded using a Phenom ProX scanning electron microscope (Thermo Fischer Scientific, Waltham, MA, USA). Prior to the measurements, samples were coated with a gold layer with a thickness of 4 nm. Images were recorded at an acceleration voltage of 10 kV. The shape of particles is defined by visual observation. 

#### 2.2.2. Laser Diffraction 

Particle size distributions (PSD) were determined (n = 3) by dry laser diffraction (Helos/KR, Sympatec, Clausthal-Zellerfeld, Germany). A dry dispersion unit with a feed rate of 50% and an air pressure of 0.5 bar was used. The particle size was reported as a volume equivalent sphere diameter. The 10%, 50%, and 90% cumulative undersize of the volumetric distribution was described as ×10, ×50, and ×90 respectively. The span of the volumetric particle size indicates the width of the particle size distribution and was calculated from the indicated parameters with the equation
Span = (×90 − ×10)/×50(1)

#### 2.2.3. Moisture Content

The total moisture content (KF) was determined (n = 2) via Karl-Fisher titration. The loss on drying (LOD) is determined (n = 2) with a Sartorius MA150 Q moisture analyzer (Sartorius AG, Göttingen, Germany). Samples are measured according to the Ph. Eur. Methods. 

#### 2.2.4. Specific Surface Area

Specific surface area (SSA) is determined (n = 2) with a Tristar II physisorption instrument (Micromeritics, Norcross, USA) based on a static volumetric technology. Samples are degassed at 40 °C for 2 h under nitrogen flow before analysis. Krypton is used as adsorption gas to analyze samples of 1–2 g. Isotherm data were elaborated with Tristar II 3020 2.02 software (Micromeritics, Norcross, GA, USA) using a Brunauer–Emmett–Teller analysis (BET) model. 

#### 2.2.5. Bulk and Tapped Density

Bulk and tapped density were measured (n = 2) according to Ph. Eur. Method 1. 100 g of powder was poured into a 250 mL graduated cylinder mounted on an automatic tapping device STAV 2003 stampfvolumeter (Engelsmann, Ludwigshafen am Rhein, Germany). The Hausner ratio (HR) was calculated as the quotient of the tapped density (TD) and the bulk density (BD)
HR = TD/BD(2)

#### 2.2.6. Ring Shear Testing

A ring shear tester (RST-XS, Dietmar Schulze, Wolfenbuttel, Germany) was used to measure (*n* = 2) the flow function coefficient (ffc). The ffc is defined as the ratio of the consolidation stress and the unconfined yield strength. Powders were measured at a pre-consolidation stress (σ_pre_) of 4 kPa and normal stresses of 1, 2, and 3 kPa were used for shear to failure.

#### 2.2.7. Charge Density

Triboelectric charging of powders was measured (*n* = 3) with a GranuCharge (GranuTools, Awans, Belgium). The initial charge was measured by introducing the powder inside the Faraday cup connected to an electrometer. The initial charge density (q_0_) was calculated by dividing the net charge by the mass of the powder sample. After this measurement, powders were fed into V-shaped stainless-steel 316 L tubing using a vibratory feeder. At the end of the tubing system, samples were collected inside a Faraday cup connected to an electrometer to determine the final charge density (q_f_). The tribo-charging density variation (Δq) was calculated as the difference between final (q_f_) and initial charge density (q_0_)
Δq = q_f_ − q_0_(3)

#### 2.2.8. True Density and Heckel Testing

The true density (TrD) of samples was determined (*n* = 2) with an AccuPyc II 1340 Helium pycnometer (Micromeritics, Norcross, USA). The equilibrium rate was 0.02 psig and the number of purges 5. 

A compaction simulator (Phoenix, Brierley Hill, UK) with a V-shaped compaction profile and 10 mm round flat faced punches was used to perform Heckel analysis (*n* = 3). The punch speed for the slow evaluation was set to 0.01 mm/s and the fast at 300 mm/s. Data were analyzed by the compaction analysis software program to generate values for yield pressure (P_y_) using the Heckel equation with the relative density of the compact (D), the applied pressure (P), and the gradient of the line in the linear region (k)
ln(1/(1 − D)) = kP + A(4)

The strain rate sensitivity (SRS) is calculated by comparing the yield pressure at high speed (PyF) and slow speed (PyS)
%SRS = (P_y_F − P_y_S)/(P_y_S) × 100(5)

### 2.3. Preparation of the Blends

A fine lactose grade (Pharmatose^®^ 450 M) is blended with a coarse sieved lactose (Pharmatose^®^ 80 M) in a Turbula blender T2 at 96 rpm for 8 min. Amounts of fines were increased in steps of 5% w/w between 0% and 40% w/w fines, and in steps of 20% w/w between 40% and 100% w/w fines. 

### 2.4. Rotating Drum Method

The rheological behavior of different excipient grades and blends is evaluated (*n* = 2 (Note that blends with 5% w/w, 10% w/w, 15% w/w, 25% w/w, and 40% w/w fines were measured once (n = 1) instead of in duplicate. Replicates on other batches ensure robustness and repeatability of measurements)) with the rotating drum method GranuDrum (GranuTools, Awans, Belgium). Rotational speed is increased from 2–20 rpm in steps of 2 and from 20–60 rpm in steps of 5 rpm. For each rotating speed, 40 snapshots of the powder bed separated by 1 s are taken by a CCD camera. The position of the powder/air interface in these snapshots is detected by an edge detection algorithm. The average interface position is used to compute the flowing angle. The standard deviation from the temporal fluctuations of the interface is used to compute the dynamic cohesive index. A cohesive powder leads to an intermitted flow while a non-cohesive powder leads to a regular flow. Therefore, a dynamic cohesive index close to zero corresponds to a non-cohesive powder. When the powder cohesiveness increases, the cohesive index increases accordingly. The cohesive index measure has been shown in previous work to be relevant to evaluate the powder cohesiveness [23,24].

A rheological index parameter (RI) is defined as the slope of a linear fit of the dynamic cohesive index as function of rotational speed. Positive rheological index indicates shear thickening, while negative rheological index indicates shear thinning. Powder that exhibits shear thickening behavior will have increased cohesiveness at higher stresses, which is associated with a decrease of flowability. Shear thinning indicates opposite behavior. 

### 2.5. Multivariate Analyses

Partial least squares (PLS) type MVA models were developed with Simca-P 16 software (Umetris, Umeå, Sweden). Models were developed by regressing the material property descriptors in Table 1 (X) versus the Rheological Index parameter (Y). The importance of X-variables is evaluated by a variable influence on projection (VIP). Values for the material property descriptors that are used to create PLS models are indicated in Appendix A.

### 2.6. Tableting

For the formulations that are tableted, 99.5% w/w filler is blended with 0.5 % w/w MgSt for 2 min in a Turbula blender T2 at 96 rpm. Blends are compressed on a RoTab rotary tableting press at 25 rpm. 9 mm flat beveled punches (iHolland) are used and compaction force is set to 10 kN. The filling depth is set to obtain tablets of 250 mg at 10 rpm agitator (optfiller) speed. The agitator speed is increased from 10–45 rpm in steps of 5 rpm to get different amounts of agitation without changing any further settings. 

### 2.7. Tablet Testing

Tablets are analyzed on weight by using an automated tablet tester (Sotax AT50). Twenty tablets are analyzed and the average and standard deviation is reported.

## 3. Results and Discussion

### 3.1. Raw Material Characterization

A range of physical properties of the materials that are evaluated in this study are shown in Table 2. SEM pictures that are used to identify the shape of particles are provided in Appendix A. Physical properties shown are expected to be relevant for flow behavior, although a broader set of material properties was used to characterize the materials and to investigate correlations with the Rheological Index in the PCA. Additional parameters that are measured for all materials are provided in Appendix A. 

The used set of materials covers a large variation in excipient type, particle shape, particle size distribution, density, and powder flow parameters. Four grades (4) of anhydrous lactose are evaluated. SuperTab^®^ 21 AN and Lactopress^®^ anhydrous have a relatively large proportion of fines, as indicated by the ×10 of 15 μm. SuperTab^®^ 22 AN contains a smaller proportion of fines and has improved flow properties according to the Hausner ratio and the flow function coefficient. SuperTab^®^ 24 AN has also improved flow properties and a slightly different shape and lower density. Four grades of non-modified lactose monohydrate are evaluated. Pharmatose^®^ 80 M is a sieved tomahawk shaped material, with a relatively large particle size. Pharmatose^®^ 150 M, Pharmatose^®^ 200 M and Pharmatose^®^ 450 M are milled materials. Particle size, density, and flow properties of these grades decrease with increasing number. The four evaluated modified lactose monohydrate grades consist of three spray dried grades with slightly different particle size distribution and density, and one granulated lactose monohydrate. Flow parameters of these grades all indicate very good flowability. More irregular shaped particles that are evaluated are microcrystalline cellulose and Primellose^®^. Pharmacel^®^ 101 and Pharmacel^®^ 102 are two microcrystalline cellulose grades with spherical morphology consisting of fibers. Particle size of Pharmacel^®^ 101 is smaller than for Pharmacel^®^ 102, which also explains the reduced flow properties. Pharmacel^®^ sMCC90 is very similar to Pharmacel^®^ 102, but has been co-processed with 2% w/w silicon dioxide to increase the specific surface area and improve the flow properties. Primojel^®^ and Primellose^®^ are two superdisintegrants with small particle size and a spherical and fibrous morphology respectively. The good flow indicated by the flow parameters of Primojel^®^, is in line with the expectation from the spherical morphology. 

Particle size distribution, particle shape and density are parameters that are known to have an impact on the different forces acting on particles during powder flow. This can be understood by looking at the different driving and drag forces that act on particles and determine the powder flow. One of the driving forces for flowability is gravity. Gravitational forces are higher for larger particles, and for particles with higher (true) density [25]. Drag forces on the other hand, typically include adhesive and cohesive forces, which produce a tendency for particles to stick to each other and to other surfaces. Adhesive and cohesive forces are composed mainly of van der Waals forces, capillary bridging, and electrostatics [26]. The magnitude of these forces depends on the nature of the material and on the available surface. Adhesive and cohesive forces will be higher for smaller or irregular shaped particles, as the available surface for these particles is higher [23]. Irregular shaped particles also can have a negative impact on flowability due to the increased risk for mechanical interlocking [27,28]. 

### 3.2. Cohesive Index as a Function of Rotational Speed—From Static to Dynamic Regimes

The rheological behavior of the different excipients was evaluated with the rotating drum method GranuDrum. GranuDrum measurements allow to follow the evolution of the flow properties as function of the flow field. At each rotational speed value, the flowing angle is computed from the average interface position, and the dynamic cohesive index is computed from the interface fluctuations. The dynamics of the flowing angle of non-cohesive granular materials is well-known and described in literature. In the considered range of rotating speed, non-cohesive materials lead to continuous flow and the shape of the interface is typically flat and easy to analyze [29]. However, in this paper, we also deal with cohesive powders, of which the flow is irregular and more complex [30,31]. In this paper, we therefore do not focus on the flowing angle as a flow parameter, but on the dynamic cohesive index (CI). The value of the cohesive index is close to zero for non-cohesive powders, and increases with the cohesion of the material. 

To highlight the influence of the different flow regimes, the cohesive indices at 2 rpm and 60 rpm are compared with other flow parameters Hausner ratio and ffc from shear cell testing. Figure 1 shows the cohesive index obtained by GranuDrum measurements at 2 rpm and 60 rpm rotational speed as function of ffc and Hausner ratio. At rotational speeds of 2 rpm, the powder is close to the quasi-static regime and the flow behavior is dominated by inter-particle forces, often referred to as cohesiveness [23]. The observations and classification of the GranuDrum at this speed are in line with results obtained in the (quasi-)static flow regime in previous work [32]. Within each category of products, flow ranking based on the cohesive index at 2 rpm is in line with the flow ranking according to Hausner ratio and ffc. For anhydrous lactose rotating at 2 rpm, SuperTab^®^ 21 AN and Lactopress^®^ anhydrous have the highest cohesive index (37–43), while SuperTab^®^ 22 AN and SuperTab^®^ 24 AN have low cohesive index (18). This is in line with the order indicated by Hausner ratio and the shear cell testing at 4 kPa pre-consolidation strength with ffc values >10 for SuperTab^®^ 22 AN and SuperTab^®^ 24 AN and around 7.5 for the other two grades. For lactose monohydrate samples a similar trend is observed. Pharmatose^®^ 80 M has a low cohesive index of 22 at 2 rpm rotational speed, which is in line with a low Hausner ratio and high ffc. The milled lactose monohydrate with smaller particle size all have a cohesive index above 50 at low rotational speed, which is in line with higher Hausner ratio and lower ffc. Modified lactose monohydrate grades all have a cohesive index below 20 at low rotational speeds, which is also in line with low Hausner ratio and ffc >10. For irregular shaped excipients like microcrystalline cellulose and Primellose^®^, the cohesive index is around 30. Primojel^®^ and Pharmacel^®^ sMCC90 have a cohesive index below 20, in line with the lowest Hausner ratio and highest ffc within its category. Also, over the different product groups, the variance in cohesive index at 2 rpm can be explained for over 60% by linear models of both the Hausner ratio and the flow function coefficient at 4 kPa pre-consolidation strength. The high overlap in powder ranking by these measurements is explained by similarities in the stress state and flow field during these measurements. 

At rotational speed of 60 rpm however, no significant correlation with the Hausner ratio or flow function coefficient can be drawn. Fitted linear models for Hausner ratio and flow function coefficient both explain the variance in cohesive index less than 2%. The lack of overlap in powder ranking by these measurements is explained by differences in the flow fields of these measurements. Hausner ratio and shear cell consider a quasi-static flow field [33], while the cohesive index at 60 rpm measures powder flow in a dynamic flow field. Observed differences are in line with Lumay et al. [23], who showed that different flow regimes were influenced by different material properties. In the quasi-static or plastic regime, velocities are small or zero and the space between neighboring particles is low. Powder behavior is dominated by inter-particle contact forces [30]. In the rapid flow regime, particles are moving so fast that friction can be neglected and only short collisions between particles determine the character of flow [34]. The differences in flow classification due to differences in stress state and flow field was also confirmed by other authors [1,3,11,17,23,26,30,34,35,36].

Cohesive index values as function of rotational speed for the tested materials are presented in Figure 2. First of all, it is important to mention that no irreversible rheological behavior was observed for any of the powders. This was indicated by the absence of a hysteresis between original curve, taken during the increasing rotational speed sequence (2–60 rpm) and the and the reverse curve, taken during the decreasing rotational speed sequence (60–2 rpm). The reverse curve is not shown for clarity reasons. The reversibility in the rheological behavior indicates that no irreversible changes in material properties are induced by the measurement method. 

Most powders that are evaluated show shear thickening behavior. This indicates that cohesiveness increases with rotational speed, which is associated with more intermittent and irregular flow. This can be concern for the pharmaceutical industry as more cohesive powders can lead to increased variability and mass flow excursions outside the acceptable target range, as demonstrated by Allenspach et al. [37]. Notable is the shear thinning behavior, which is observed for two grades of anhydrous lactose, two grades of lactose monohydrate and microcrystalline cellulose. The shear thinning behavior can be explained by aeration of the powder at higher rotational speeds, which increases the distance between particles and thereby reduces the cohesive surface interactions [36]. 

The observed differences in cohesive index and rheological index for sieved (Pharmatose^®^ 80 M) and milled lactose (Pharmatose^®^ 150 M and Pharmatose^®^ 200 M) are in line with findings of Hickey et al. [38] for inhalation grades lactose. These authors reported that an apparent contradiction is present in flow property parameters of milled and sieved powders. Static powder flow measurements (bulk and tapped density, angle of repose) lead to the conclusion that milled powders exhibit poor flow compared to sieved batches. Dynamic rotating drum measurements indicated the reverse, where milled powder flows better than sieved powder. 

To evaluate which material properties influence the shear thinning and thickening behavior, a new parameter rheological index (RI) is introduced. The Rheological Index (RI) is defined as the linear slope of dynamic cohesive index as function of rotational speed between 2 rpm and 60 rpm. Figure 3 shows the RI values for the investigated materials. For Lactopress^®^ anhydrous, SuperTab^®^ 21 AN, Pharmatose^®^ 150 M, Pharmatose^®^ 200 M, and Pharmacel^®^ 101 the reduction in cohesiveness due to aeration seems to outweigh the inertial effect, resulting in shear thinning behavior. For Pharmacel^®^ 102, Primojel^®^, and Primellose^®^ the rheological index is close to zero. 

### 3.3. Multivariate Analyses to Reveal Drivers for the Rheological Index 

A partial least squares (PLS) model was created to identify which material properties impact the rheological index parameter (RI) of powders the most. In the current approach, PLS was used as exploratory technique and no optimization of the model was performed. Figure 4 shows the score and loading plot of the PLS model with two principal components. In the PLS loading plot the parameters ×10 and ffc point in the same direction as the RI, indicating a positive correlation. The parameter span is indicated at the exact opposite direction, suggesting a negative correlation. In the PCA loading plot RI is pointing towards the upright direction in quadrant 1. The four modified lactose monohydrate grades and Pharmatose^®^ 80 M are located in the PCA scoring plot in quadrant 1 towards the positive direction of rheological index. Pharmatose^®^ 150 M, Lactopress^®^ anhydrous and SuperTab^®^ 21 AN are located in quadrant 3 towards the negative direction of rheological index. 

A variable influence on projection (VIP) plot is constructed to evaluate which material properties have the most significant correlation with rheological behavior. From the VIP plot in Figure 5 it can be observed that the parameters that are the most important in this model towards RI are related to the particle size distribution (×10, span) and parameters related to powder flow (ffc, HR). No relevant contribution is present for true density, yield strengths, SRS, electrostatics, tapped density, and moisture content. 

### 3.4. Variation of the Amount of Fines

To test the impact of particle size (×10 and span) on the rheological index that was indicated by the PLS, blends with different particle size distributions are further evaluated. Sieved lactose with different amounts of fines added are tested with the rotating drum method. Particle size distribution parameters are indicated in Appendix A. Figure 6 shows the result of the rotating drum measurements, from which the rheological parameter is derived and summarized in Figure 7. The results of the dynamic flow measurements show that the cohesive index at low rotational speed increases with an increased fraction of fines. This can be understood by looking at the flow regime of the powder. At low rotational speed, the powder is in the quasi-static or plastic regime and the space between neighboring particles is low. Powder behavior is dominated by inter-particle contact forces, like cohesive and adhesive forces [30]. Cohesive and adhesive forces are typically higher with more fine particles present, due to the increased availability of surface and because fines support powder packing [23,38]. At the same time, flowability driving gravitational forces that act on small particles are typically lower, due to the lower mass of these particles. 

At higher rotational speed, addition of 5–35% w/w of fines resulted in reduced cohesive index. This is also visible in the reduced rheological index when up to 35% w/w fines are added. Negative rheological index is observed for blends with 10–35% w/w fines. Notable is the tipping point between 35% w/w and 40% w/w of fines. At this point the cohesive index at 2 rpm remains constant, but the rheological index suddenly turns from negative to positive, resulting in significantly larger cohesive index at high rotational speeds. A tipping point in flow behavior around 30% w/w of fines was also observed before by Molerus and Nwylt in quasi-static shear cell flow measurements of binary mixtures of coarse and fine limestone particles [39]. They found that flow behavior at a fines content above 30% w/w was dominated by the behavior of fines. More recently, Pillitery et al. [40] also found a tipping point in compaction of binary granular mixture close to 30–35% w/w of fines. It is expected that, at this level of fines, the coarse particles are completely embedded by the fines, and so flow behavior is governed by interparticle forces between fines. Below this level, contacts between coarse particles dominate flow [41].

The observed differences in cohesive index and rheological index for blends with different amounts of fines, are in line with the earlier mentioned findings of Hickey [38] on inhalation lactose. The authors explained that milled powder flows more readily because of the presence of fines. Stretching the findings of Hickey et al. [38], the current studies shows that the rheological behavior is not only different for milled and sieved materials. It reveals that the presence of a small amount of fines in a powder with larger particle size distribution can have a positive effect on the dynamic flow properties and rheological behavior. 

An explanation for the reduced cohesive index of blends with 10–35% w/w fines at high rotational speeds is found in the flow field. In a more rapid flow regime, the magnitude of forces acting on the particles are different. On the one hand, the inertial effect is stronger, which could lead to increased cohesive index. On the other hand, friction by cohesive and adhesive forces is typically reduced. This is due to the fact that rolling friction is typically much smaller than static or sliding friction [34]. It is the authors hypothesis that the presence of fines even further reduces the friction when particles are moving. This is explained by a reduction of bridge formation in moving powder when small particles are present between larger particles. This improves particle separation, resulting in more readily powder flow. In this study, we find that blends with 10–35% w/w fines correspond to optimal mixtures where the separation of large grains by interstitial fines improves the rheological properties of the blends. Indeed, the cohesive index of the blends with 10–35% w/w fines reduces with the rotational speed, which can be explained by an aeration of the powders leading to a reduction of the cohesive forces acting on the grains. When the fines content increases over 40% w/w, the flow behavior is dominated by the fine fraction embedding the coarse particles, and a more cohesive powder is observed. The hypothesis is that for blends of coarse and fine material a tipping point for rheological behavior exist at the point where coarse particles are completely embedded by fines. This tipping point is therefore expected to depend on the particle shape and particle size ratio of the fines and the coarse particles.

### 3.5. Correlating Rheological Behavior to Tableting Performance

To evaluate the practical application of shear thickening and shear thinning, a tableting study was performed based on two excipients that showed these two types of rheological behavior. Direct compression grades were selected with a cohesive index below 50 over the entire range and strong rheological behavior. SuperTab^®^ 11 SD showed strong shear thickening with a rheological index of 0.59 rpm^−1^, while SuperTab^®^ 21 AN did show shear thinning with a rheological index of −0.48 rpm^−1^ in the GranuDrum. The mass and mass variability of tablets was evaluated as a measure for powder flow into the dies during a tableting process. The agitator speed was increased in steps of 5 rpm between 10–45 rpm to investigate different flow fields and stress states. Figure 8 shows the average mass and the variability in the mass of tablets that was obtained for tableting with different agitator speeds. 

Differences in rheological behavior were also visible during flow into the dies in the tableting process. Increased agitator speed resulted in higher mass tablets with less variability for the formulation with SuperTab^®^ 21 AN. This indicates that at low agitator speed, the powder flow into the dies was incomplete and that improvement was obtained by increased flow via adjustment of the flow field. These findings confirm that powders with a negative rheological index exhibit a lower cohesion and better flowability with a more dynamic process. This is especially visible for grades like SuperTab^®^ 21 AN, which have as a significant fraction of fines present and a large bridging propensity in the quasi-static state [32]. For SuperTab^®^ 11 SD on the other hand, increased agitator speed did not result in higher mass tablets. A slight increase in mass RSD is observed for this formulation, indicating more cohesive powder flow. This is in line with the positive rheological index of SuperTab^®^ 11 SD, in line with the sharp particle size distribution and low bridging propensity in the quasi-static state [32]. These results show that rheological index measurements in the lab can be a tool to set-up a tableting process. 

## 4. Conclusions

In this study, the effect of rheological behavior during powder flow into tablet dies is investigated. It was shown that a difference in flow field and stress state can have an effect on the cohesive index and die filling of powders. This effect depends on the raw material properties. A new parameter for rheological behavior (RI) was introduced, which is a measure for the change in dynamic cohesive index upon changes in stress state and flow field. Most powders evaluated show shear thickening behavior. However, shear thinning is observed for some materials as well. Of all physical/chemical parameters tested, the particle size distribution (×10) was shown to have the largest impact on the rheological behavior of powders. At higher rotational speeds, addition of 5–35% w/w of fines to coarse sieved lactose resulted in a decrease of cohesive index. This is explained by the presence of fines between larger particles, that can improve particle separation thus reducing the cohesive forces acting between grains. The fidelity of the rheological index was validated by correlating it to tableting performance in a rotary tablet press with agitators. It was shown that insights on the rheological index (RI) obtained by rotating drum experiments can be used to optimize agitator settings in a tableting machine. 

## Figures and Tables

**Figure 1 pharmaceutics-13-01198-f001:**
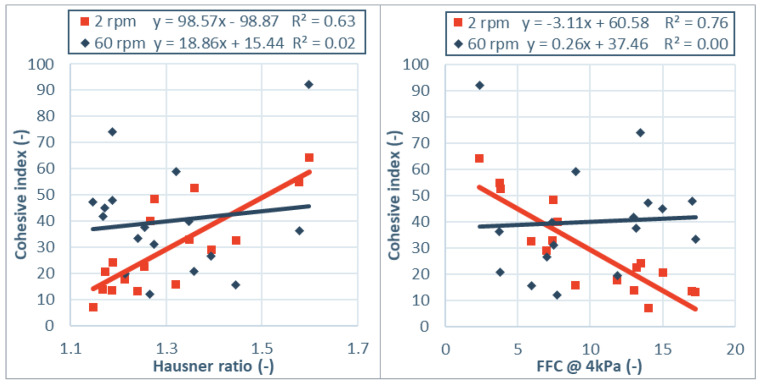
Cohesive index values as function of Hausner ratio (**left**) and flow function coefficient (**right**) at two different rotational speeds. The obtained cohesive index at low rotational speed (2 rpm) correlated with both measurements, while no correlation is observed for the cohesive index at high rotation speed (60 rpm).

**Figure 2 pharmaceutics-13-01198-f002:**
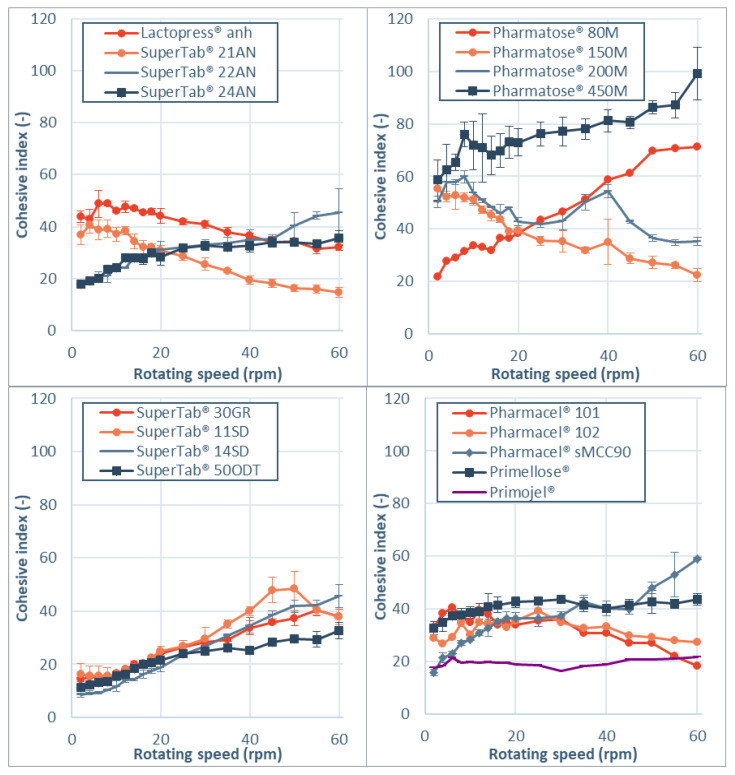
Cohesive index values as function of rotational speed for anhydrous lactose (**top left**), lactose monohydrate (**top right**), modified lactose monohydrate (**bottom left**) and microcrystalline cellulose/superdisintegrants (**bottom right**).

**Figure 3 pharmaceutics-13-01198-f003:**
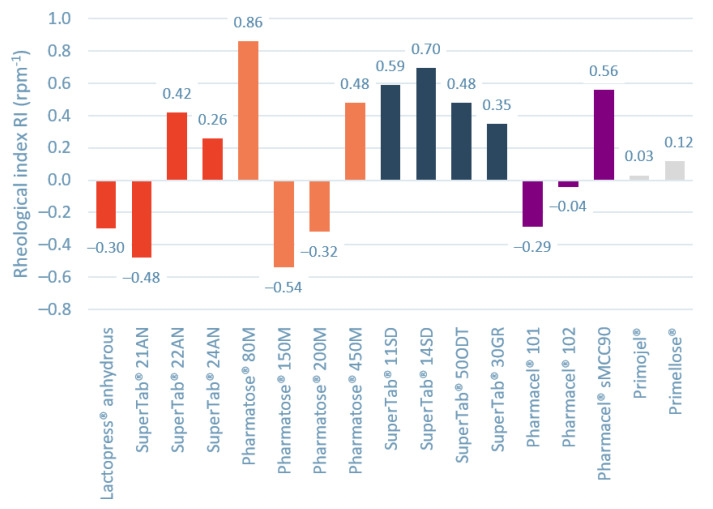
Rheological index (RI) values for the different materials. A positive rheological index indicates shear thickening behavior, while negative rheological index indicates shear thinning behavior. Different colours represent different types of materials.

**Figure 4 pharmaceutics-13-01198-f004:**
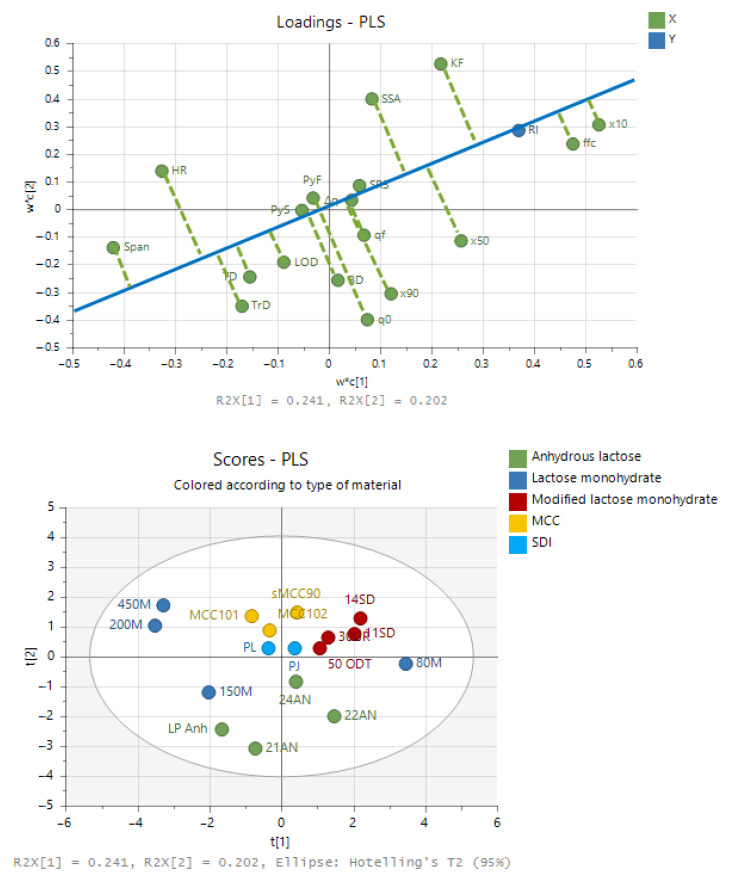
PLS analyses of the dataset with a loading plot (**top**) and score plot (**bottom**).

**Figure 5 pharmaceutics-13-01198-f005:**
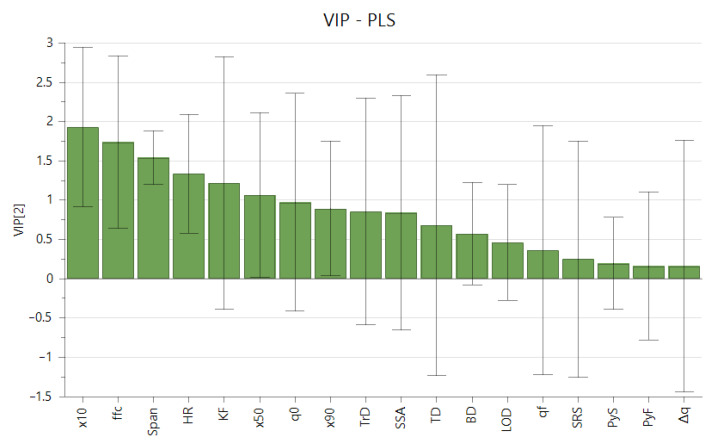
Variable Influence on Projection (VIP) plot indicating the relevance of terms for explaining the rheological index. Error bars indicate the 95% confidence interval.

**Figure 6 pharmaceutics-13-01198-f006:**
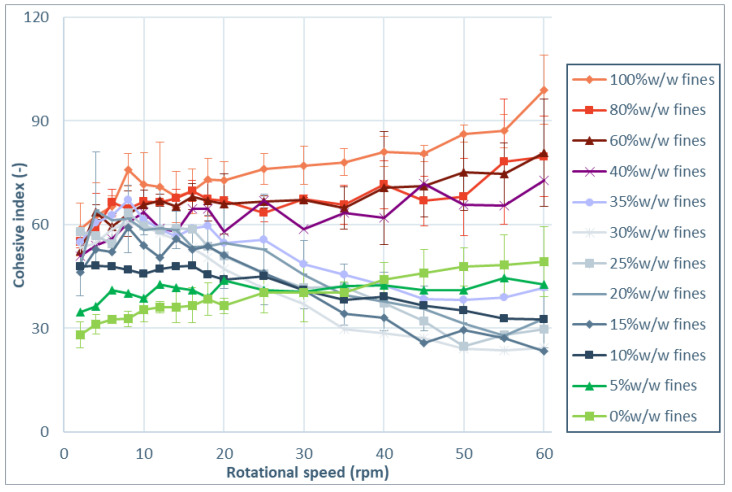
Cohesive index as function of rotational speed for sieved lactose with different amounts of fines added.

**Figure 7 pharmaceutics-13-01198-f007:**
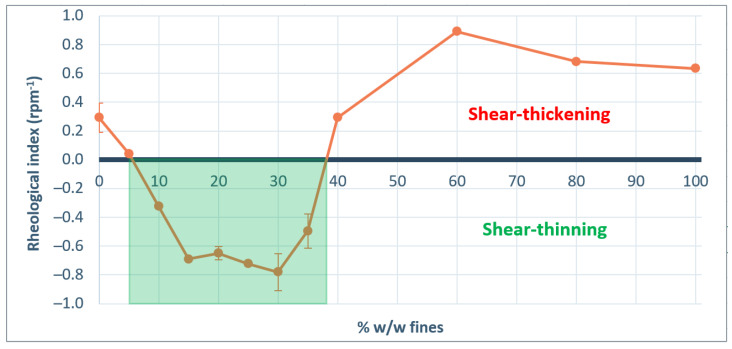
Rheological index parameter as function of the amount of fines added to the sieved lactose. The marked area at 10–35% w/w fines indicates shear thinning behavior.

**Figure 8 pharmaceutics-13-01198-f008:**
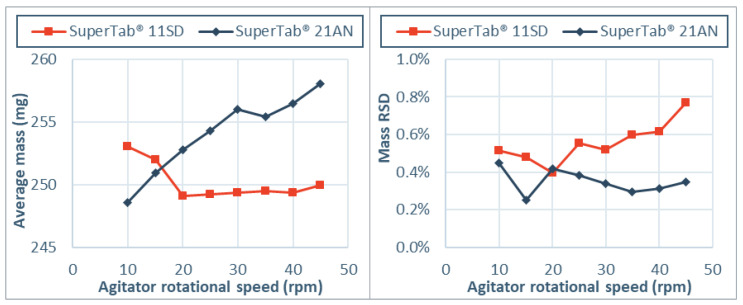
Tableting results (n = 20) on average mass and mass variability (%RSD) of a shear thinning material (SuperTab^®^ 21 AN) and a shear thickening material (SuperTab^®^ 11 SD). The agitator rotational speed was increased from 10–45 rpm in steps of 5 rpm.

**Table 1 pharmaceutics-13-01198-t001:** List of material characterization techniques, the material properties they measure and corresponding abbreviations of the measured material properties

Characterization Technique	Physical Property	Abbreviation	Range of Values	Unit
Visual observation by scanning electron microscopy	Shape	Shape	-	-
Particle size distribution (PSD) by laser diffraction	10% cumulative undersize of volumetric PSD	×10	3.0–77.6	μm
	50% cumulative undersize of volumetric PSD	×50	18.3–243	μm
	90% cumulative undersize of volumetric PSD	×90	49.4–406	μm
	Span of the volumetric PSD*	Span	1.22–2.83	-
Karl fisher titration	Total moisture content	KF	0.1–5.8	%w/w
Thermogravimetric balance	Loss on drying	LOD	0.0–9.3	%w/w
Brunauer–Emmett–Teller analysis with Krypton	Specific surface area	SSA	0.1–5.1	m^2^/g
Graduated cylinder	Bulk density	BD	0.35–0.79	g/mL
	Tapped density	TD	0.49–0.98	g/mL
	Hausner ratio	HR	1.15–1.60	-
Ring shear cell tester	Flow function coefficient at 4 kPa pre-consolidation pressure	ffc	2.4–17.3	-
Electric charge analyzer	Initial charge density	q0	−1.5–0.2	nC/g
	Final charge density	qf	−1.5–0.2	nC/g
	Tribo-charging density variation	Δq	−5.0–0.1	nC/g
Gas pycnometer	True density	TrD	1.52–1.58	g/mL
Heckel testing by compaction simulation	Yield pressure at 0.01 mm/s—slow	PyS	78–229	MPa
	Yield pressure at 300 mm/s—fast	PyF	80–236	MPa
	Strain Rate Sensitivity	SRS	0–48	%
Rotating drum	Rheological index	RI	−0.54–0.86	rpm^−1^

**Table 2 pharmaceutics-13-01198-t002:** Physical properties such as type, shape and particle size distribution, density and flow properties for the set of excipients that is used in this study

Grade	Abbreviation	Type	Shape	×10 (µm)	×50 (µm)	×90 (µm)	Span	Bulk Density (g/mL)	Hausner Ratio (-)	ffc @4 kPa (-)
Lactopress^®^ anhydrous	LP anh	Anhydrous lactose	Shards	16.5	133	323	2.30	0.69	1.28	7.5
SuperTab^®^ 21 AN	21 AN	Anhydrous lactose	Shards	24.1	180	387	2.02	0.72	1.27	7.7
SuperTab^®^ 22 AN	22 AN	Anhydrous lactose	Shards	47.0	203	359	1.54	0.68	1.17	15
SuperTab^®^ 24 AN	24 AN	(Granulated) anhydrous lactose	Granular	37.0	121	298	2.15	0.54	1.25	13
Pharmatose^®^ 80 M	80 M	Lactose monohydrate (sieved)	Tomahawk	76.6	242	406	1.36	0.79	1.19	13
Pharmatose^®^ 150 M	150 M	Lactose monohydrate (milled)	Tomahawk/fines	7.4	68.4	189	2.66	0.72	1.36	3.8
Pharmatose^®^ 200 M	200 M	Lactose monohydrate (milled)	Tomahawk/fines	4.4	37.7	111	2.83	0.62	1.58	3.7
Pharmatose^®^ 450 M	450 M	Lactose monohydrate (milled)	Fines	3.0	18.3	49.4	2.54	0.50	1.60	2.4
SuperTab^®^ 30 GR	30 GR	Modified lactose monohydrate	Granular	38.3	126	297	2.05	0.63	1.24	17
SuperTab^®^ 11 SD	11 SD	Modified lactose monohydrate	Spherical	44.0	119	223	1.51	0.63	1.19	17
SuperTab^®^ 14 SD	14 SD	Modified lactose monohydrate	Spherical	47.7	124	227	1.44	0.62	1.15	14
SuperTab^®^ 50 ODT	50 ODT	Modified lactose monohydrate	Spherical	30.9	106	199	1.58	0.71	1.17	13
Pharmacel^®^ 101	MCC101	Microcrystalline cellulose	Spherical/Fibers	20.0	62.2	137	1.89	0.34	1.45	5.9
Pharmacel^®^ 102	MCC102	Microcrystalline cellulose	Spherical/Fibers	29.9	86.9	200	1.95	0.33	1.39	7.0
Pharmacel^®^ sMCC90	sMCC90	Microcrystalline cellulose, co-processed with silicon dioxide	Spherical/Fibers	29.4	102	233	1.99	0.38	1.32	9.0
Primojel^®^	PJ	Superdisintegrant	Spherical	21.1	42.3	72.6	1.22	0.79	1.21	12
Primellose^®^	PL	Superdisintegrant	Fibers	24.4	54.4	114	1.65	0.55	1.35	7.4

## Data Availability

Not applicable.

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
