# Peer review of "Impact of Powder Properties on the Rheological Behavior of Excipients"

_pharmaceutics, 2021, doi:10.3390/pharmaceutics13081198_

Round 1

Reviewer 1 Report

Hello Authors,

Please see my comments below,

1) Why are some parameters measured in triplicate and others in duplicate?

2) Few instances in the manuscript have the following "Error! Reference source not found."

3) What method was used to measure the shape of the materials?

4) Include how the dynamic cohesive index was measured in the Granudrum instrument.

4) Authors mention that the shear cell testing considers a static flow field. However, there is shear happening on the powder bed to create the failure, so there is a dynamic aspect to it. Same is true for angle of repose, where it flows through the funnel.

5) It is not very clearly defined, what do the authors mean by shear thickening and/or thinning effect when it comes to powders. 

6) Without fully explaining the principles and procedures of measuring the dynamic cohesive index, the conclusions drawn between RI values and aeration, inertial effect, shear thinning/thickening of powders is not clearly understood.

7) Correlating Rheological Behavior to Tableting Performance: The data presented in this section does not fully support the conclusions made. Super tab 11sd has a negative RI. Why would it show higher weight variation at a higher agitation speeds ?

8) In the methods section, there is a talk about true density, heckel analysis using compaction simulator. I don't see that data though.

Author Response

We thank the reviewer for evaluation of the manuscript and for the challenges on the content. The revisions are addressed in the detailed comments in the attachement. 

Reviewer 2 Report

The authors provide a comparative study of rheological properties of different excipients of different nature. They compare some of the index that characterize flowability, finding correlations only with low shear speeds, for which they pose a hypothesis already raised by other authors in previous works. A multivariate analysis is carried out focused on detecting the critical factors involved especially in the rheological index. They study how the cohesive index changes when the% of fines in a mixture increase. The study reveals the interest of the rheological index, mainly and its utility to optimize agitator settings in a tableting machine. In my opinion the study is correct and interesting because the index proposed could be useful in QbD approach. There are some aspects to be enhanced.

  1. Table 1, located in methodology part, show results. This table is a resume of major part of the tests. Should be ok if these primary data are shown in supplementary files.
  2. Results presented for Karl Fischer titration have no replicates, so there are no SD.
  3. Line 170 and others among the text: error with link to the source (figures or tables).
  4. Table 2 it is no mentioned in the text.
  5. Figure 1 name is used for two graphs.
  6. Line 287. Where authors refer to Figure 1, maybe they want to say Figure 2
  7. Figure 3 caption should be improved in order to clarify the figure.
  8. Line 347. The authors say that there is no significant contribution for several factors. Is it possible to show the p values? Did the authors study if any factors included in this correlation analysis is a covariable?
  9. Caption Figure 6 should be improved explaining what the coloured region represents.

Author Response

We thank the reviewer for the evaluation of the manuscript. The proposed revisions are addressed in the detailed comments in the attachement.

Reviewer 3 Report

The overall manuscript flows well and can be useful for formulators.

Line:170: "Models were developed by regressing the material property descriptors in in Error! Reference source not found." Not clear about this sentence

The authors explain the sudden tipping in the Cohesive Index at higher speeds between 35%to 40% fines. Would this be consistent across all particle shapes, also what would be the approximate ratio of particle sizes between fines and coarse particles to hold the hypothesis?

Fig.1. Why does the Cohesive Index of SuperTab11SD increase and then decreases at higher RPM? The other lactose grades do not show similar patterns.

Studies show 30% w/w fines with the lowest cohesive index at high RPM. Would the authors elaborate on why would that be. Also, would this be useful during formulation with an Active, as high RPM may lead to eventual segregation in the blend

Author Response

(The authors gave the same response as above.)

Round 2

Reviewer 1 Report

Hello Authors,

Please see my comments to your revisions,

1) Why are some parameters measured in triplicate and others in duplicate?
Author Reply: The amount of replicates that are performed for each measurement depends on the expected variation of the method. For each parameter at least a duplicate is performed, and particle size and electrostatics are measured in triplicate to ensure proper precision of the final results.
Reviewer Reply: How do the authors know about expected variation? Is there a reference for that? 

8) In the methods section, there is a talk about true density, heckel analysis using compaction simulator. I don't see that data though.
Author Reply: The reviewer is correct that this information was missing in the manuscript. Data for the missing parameters is added in Supplementary Table 1. A reference to this table is included in the beginning of section 3.1.

Reviewer Reply: Why is Table 1 in supplementary file, and not in the manuscript? There should be a footnote below the table, for abbreviated terms in Table 1. And these parameters are not discussed at all in the "results & discussion" section. Why were these done, if there is no discussion involved?

Round 3

Reviewer 1 Report

Comments have been addressed.